# MASKED DISTILLATION WITH RECEPTIVE TOKENS

**Tao Huang**[1,2*]   **Yuan Zhang**[3*]   **Shan You**[2†]
**Fei Wang**[4]   **Chen Qian**[2]   **Jian Cao**[3]   **Chang Xu**[1]
[1]School of Computer Science, Faculty of Engineering, The University of Sydney
[2]SenseTime Research   [3]School of Software and Microelectronics, Peking University
[4]University of Science and Technology of China

## ABSTRACT

Distilling from the feature maps can be fairly effective for dense prediction tasks since both the feature discriminability and localization priors can be well transferred. However, not every pixel contributes equally to the performance, and a good student should learn from what really matters to the teacher. In this paper, we introduce a learnable embedding dubbed *receptive token* to localize those pixels of interests (PoIs) in the feature map, with a distillation mask generated via pixel-wise attention. Then the distillation will be performed on the mask via pixel-wise reconstruction. In this way, a distillation mask actually indicates a pattern of pixel dependencies within feature maps of teacher. We thus adopt multiple receptive tokens to investigate more sophisticated and informative pixel dependencies to further enhance the distillation. To obtain a group of masks, the receptive tokens are learned via the regular task loss but with teacher fixed, and we also leverage a Dice loss to enrich the diversity of learned masks. Our method dubbed MasKD is simple and practical, and needs no priors of tasks in application. Experiments show that our MasKD can achieve state-of-the-art performance consistently on object detection and semantic segmentation benchmarks. Code is available at `https://github.com/hunto/MasKD`.

## 1 INTRODUCTION

Recent deep learning models tend to grow deeper and wider for ultimate performance (He et al., 2016; Xie et al., 2017; Li et al., 2019). However, with the limitations of computational and memory resources, such huge models are clumsy and inefficient to deploy on edge devices. As a friendly solution, knowledge distillation (KD) (Hinton et al., 2015; Romero et al., 2014) has been proposed to transfer knowledge in the heavy model (teacher) to a small model (student). Nevertheless, applying KD on dense prediction tasks such as object detection and semantic segmentation sometimes cannot achieve significant improvements as expected. For example, Fitnet (Romero et al., 2014) mimics the feature maps of teacher element-wisely but it has only minor improvement in object detection[1].

Therefore, feature reconstruction for all pixels may not be a good option for dense prediction, since not every pixel contributes equally to the performance. Many followups (Li et al., 2017; Wang et al., 2019; Sun et al., 2020; Guo et al., 2021) thus dedicated to show that distillation on sampled valuable regions could achieve noticeable improvements over the simple baseline methods. For example, Mimicking (Li et al., 2017) distills the positive regions proposed by region proposal network (RPN) of the student; FGFI (Wang et al., 2019) and TADF (Sun et al., 2020) imitate valuable regions near the foreground boxes; Defeat (Guo et al., 2021) uses ground-truth bounding boxes to balance the loss weights of foreground and background distillations; GID (Dai et al., 2021) selects valuable regions according to the outputs of teacher and student. These methods all rely on the priors of bounding boxes; however, *are all pixels inside the bounding boxes necessarily valuable for distillation?*

The answer might be negative. As shown in Figure 1, the activated regions inside each object box are much smaller than the boxes. Also, different layers, even different strides of features in FPN,

---

*Equal contributions. †Correspondence to: Shan You <`youshan@sensetime.com`>.

[1]Fitnet (Romero et al., 2014) improves *Faster RCNN-R50* by only 0.5%, while has no gain on *RetinaNet-R50* (see Table 1).

| Annotations | Stage 0 | Stage 1 | Stage 2 | Stage 3 | Stage 4 |
| --- | --- | --- | --- | --- | --- |

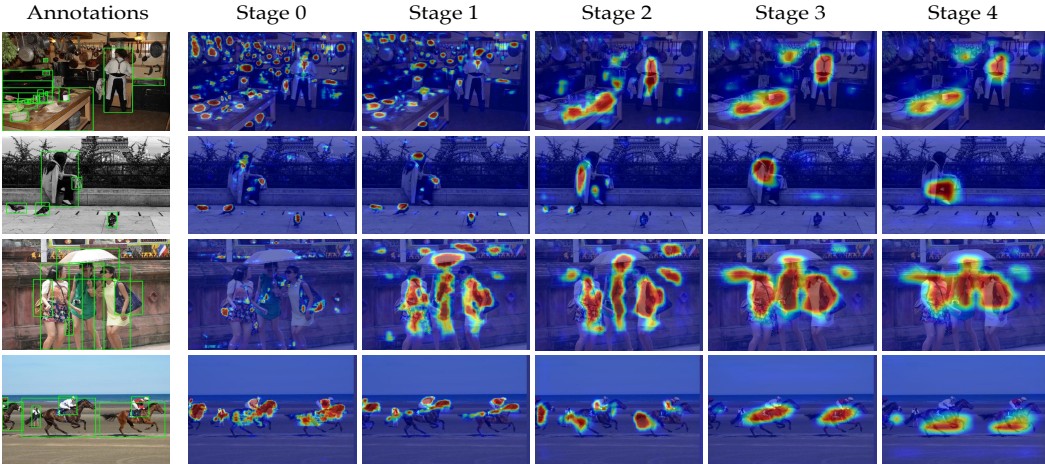

Figure 1: **Visualization of learned masks on COCO dataset.** In *Faster RCNN-R101* model, the earlier stages in FPN focus more on small objects, while the later ones focus on larger objects. Complete visualization results can be found in Appendix A.6. Zoom up to view better.

have different regions of interest. Moreover, objects that do not exist in ground-truth annotations would be treated as "background", but they actually contain valuable discriminative information. This inspires us that we should **discard the ground truth boxes and select distillation regions on a fine-grained pixel level.**

In this way, we propose to learn a pixel-wise mask as an indicator for the feature distillation. An intuitive idea is that we need to localize what pixel in the teacher's feature map is really meaningful to the task. To this end, we introduce a learnable embedding dubbed *receptive token* to perceive each pixel via attention calculation. Then a mask is generated to indicate the pixels of interests (PoIs) encoded by a receptive token. As there may be sophisticated pixel dependencies within feature maps, we thus leverage multiple receptive tokens in practice to enhance the distillation. The receptive tokens as well as the corresponding masks can be trained with the regular task loss with the teacher fixed. For the group of masks, we adopt Dice loss to ensure their diversity, and devise a mask weighting module to accommodate the different importance of masks. During distillation, we also propose to customize the learned masks using the student's feature, which helps our distillation focus more on the pixels that teachers and students really care about simultaneously.

Our MasKD is simple and practical, and does not need task prior for designing masks, which is friendly for various dense prediction tasks. Extensive experiments show that, our MasKD achieves state-of-the-art performance consistently on object detection and semantic segmentation tasks. For example, MasKD significantly improves 2.4 AP over the Faster RCNN-R50 student on object detection, while 2.79 mIoU over the DeepLabV3-R18 student on semantic segmentation.

## 2 RELATED WORK

**Knowledge distillation on object detection.** Knowledge distillation methods on object detection task have been demonstrated successful in improving the light-weight compact detection networks with the guidance of larger teachers. The distillation methods can be divided into response-based and feature-based methods according to their distillation inputs. Response-based methods (Hinton et al., 2015; Chen et al., 2017; Li et al., 2017) perform distillation on the predictions (*e.g.*, classification scores and bounding box regressions) of teacher and student. In contrast, feature-based methods (Romero et al., 2014; Wang et al., 2019; Guo et al., 2021) are more popular as they can distill both recognition and localization information in the intermediate feature maps. Unlike the classification tasks, the distillation losses in detection tasks will encounter an extreme imbalance between positive and negative instances. To alleviate this issue, some methods (Wang et al., 2019; Sun et al., 2020; Dai et al., 2021; Guo et al., 2021; Yang et al., 2021) propose to distill the features on various sophisticatedly-selected sub-regions of the feature map. For instance, FGFI (Wang et al., 2019) selects anchors overlapping with the ground-truth object anchors as distillation regions;

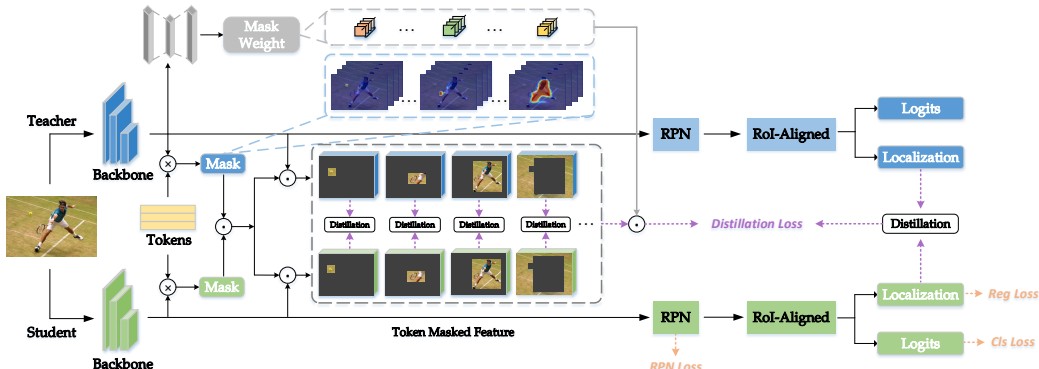

Figure 2: **Overview of our KD framework on Faster R-CNN.** We perform our masked distillation on the feature pyramid, where a set of receptive tokens are proposed to generate the masks, and a mask weighting module is conducted to adapt the loss weights for each mask. $\odot$ denotes Hadamard product, and $\otimes$ denotes matrix multiplication.

GIDs (Dai et al., 2021) distills student detectors based on discriminative instances selected by the predictions of teacher and student; In this paper, we propose to learn the pixels of interests in the feature map without box priors, and perform feature distillations on a finer pixel level.

**Knowledge distillation on semantic segmentation.** Knowledge distillation methods on semantic segmentation often focus on preserving the structural semantic relations between teacher and student. He et al. (He et al., 2019) optimize the feature similarity in a transferred latent space using a pre-trained autoencoder to alleviate the inconsistency between the features of teacher and student. SKD (Liu et al., 2019) performs a pairwise distillation among pixels to retain the pixel relations and an adversarial distillation on score maps to distill holistic knowledge. IFVD (Wang et al., 2020) transfers the intra-class feature variation from teacher to student for more robust relations with class-wise prototypes. CWD (Shu et al., 2021) proposes channel-wise distillation for a better mimic on the spatial scores along the channel dimension. CIRKD (Yang et al., 2022) proposes to learn better semantic relations from the teacher by adopting intra-image and cross-image relational distillations.

## 3 DISTILLATION WITH FEATURE RECONSTRUCTION

In order to distill the feature maps of teacher, one typical manner is to mimic the tensor pixel-wisely (Romero et al., 2014; Chen et al., 2017). Formally, with the feature maps $\boldsymbol{F}^{(t)} \in \mathbb{R}^{C \times H \times W}$ and $\boldsymbol{F}^{(s)} \in \mathbb{R}^{C_s \times H \times W}$ of teacher and student networks, where $C, H, W$ denote the number of channels, height, and width, respectively, the mimicking can be fulfilled via feature reconstruction as

$$\mathcal{L}_{\mathrm{mimic}} = \frac{1}{HWC} \left\| \boldsymbol{F}^{(t)} - \phi(\boldsymbol{F}^{(s)}) \right\|_2^2, \tag{1}$$

where $\phi$ is a linear projection layer to adapt $\boldsymbol{F}^{(s)}$ to the same resolution as $\boldsymbol{F}^{(t)}$. For example, the mimic loss $\mathcal{L}_{\mathrm{mimic}}$ is usually conducted on the outputs of feature pyramid network (Lin et al., 2017a) (FPN) for detection tasks.

However, on dense prediction tasks, the predictions are highly determined by its corresponding spatial regions in the feature maps. Treating all regions equally would weaken student's attentions on those small but critical regions, *e.g.*, the feature map contains both smaller objects and larger ones, while the larger ones obtain larger loss values as they have larger areas. Moreover, different regions often have different importance to the predictions, *e.g.*, foreground is usually more important than background in detection. Recklessly imitating the unimportant noise features would also limit the distillation performance.

As a result, a typical improvement of feature distillation in recent methods (Wang et al., 2019; Sun et al., 2020; Guo et al., 2021) is to reconstruct the features on selected and separate regions (masks). Specifically, suppose we have a set of $K$ generated masks $\boldsymbol{M} \in \mathbb{R}^{K \times H \times W}$, these methods generate

$K$ masked features and perform distillation separately, *i.e.*,

$$\mathcal{L}_{\text{feature}} = \frac{1}{K} \sum_{i=1}^{K} \frac{1}{C \sum_{j=1}^{H \times W} M_{i,j}} \left\| \boldsymbol{M}_i \odot \boldsymbol{F}^{(t)} - \boldsymbol{M}_i \odot \phi(\boldsymbol{F}^{(s)}) \right\|_2^2. \tag{2}$$

However, the masks in these methods are usually generated with the priors of bounding boxes, which are highly influenced by the annotations in ground-truth or the configurations of anchors. For example, the unlabeled boxes in the annotation data (see Figure 1) or unsuitable scales of anchors would assign wrong classes (*e.g.*, foreground and background) to some pixels, resulting in overlooked or over-valued pixels, thus weaken the performance. In this paper, we propose a novel mask generation method to locate the finer pixel-level interest areas, with neither ground-truth annotations nor predictions used.

## 4   PROPOSED APPROACH: MASKD

Our MasKD comprises two stages: the mask learning and the conventional knowledge distillation stages. In the mask learning stage, we learn the masks from a trained teacher, then adopt the learned masks to the knowledge distillation stage. Note that these two stages can be merged into one stages like previous KD methods, and the cost of the first mask learning stage is negligible. In this paper, we formulate them separately for better understanding.

### 4.1   LEARNING MASKS WITH RECEPTIVE TOKENS

We introduce a learnable embedding $\boldsymbol{E} \in \mathbb{R}^{T \times C}$ dubbed *mask tokens* to represent $T$ pixel dependencies in the feature map, then the pixels of interests (PoIs) can be obtained by calculating the similarities between mask tokens and the spatial points in the feature map:

$$\boldsymbol{M}^{(t)} = \sigma(\boldsymbol{E}\boldsymbol{F}^{(t)}), \tag{3}$$

where $\sigma$ denotes Sigmoid function, for simplicity, we flatten the teacher's feature map $\boldsymbol{F}^{(t)}$ into shape $(C, H \times W)$ in the paper, and thus the masks $\boldsymbol{M}^{(t)}$ have a shape of $(T, H \times W)$.

To learn the masks with task knowledge, we propose to train them with teacher's task loss, where a masked feature $\hat{\boldsymbol{F}}^{(t)} = \sum_{i=1}^{T} \boldsymbol{M}_i^{(t)} \odot \boldsymbol{F}^{(t)}$ is used to replace the original feature $\boldsymbol{F}^{(t)}$. Then the loss can be treated as an observation of mask quality, and we fix the teacher's weights and minimize the loss to force the masks to focus on the substantial pixels.

However, simply minimizing $\mathcal{L}_{\text{task}}^{(t)}$ would lead to an undesired collapse of the mask tokens. Specifically, some mask tokens will be learned to directly recover all the features (the learned mask is filled with 1 everywhere). However, we want to partition the feature into meaningful regions. To make the masks represent different spatial pixels, we propose a mask diversity loss based on Dice coefficient:

$$\mathcal{L}_{\text{div}} = \frac{1}{T^2} \sum_{i=1}^{T} \sum_{j=1}^{T} \rho_{\text{dice}}(\boldsymbol{M}_i^{(t)}, \boldsymbol{M}_j^{(t)}) \tag{4}$$

with

$$\rho_{\text{dice}}(\boldsymbol{a}, \boldsymbol{b}) = \frac{2 \sum_{i=1}^{N} a_i b_i}{\sum_{j=1}^{N} a_j^2 + \sum_{k=1}^{N} b_k^2}, \tag{5}$$

where $\boldsymbol{a} \in \mathbb{R}^N$ and $\boldsymbol{b} \in \mathbb{R}^N$ are two vectors. Dice coefficient $\rho_{\text{dice}}$ is widely used to measure the similarity of two images in segmentation tasks. By minimizing the coefficients of each mask pair, we can make masks associated with different PoIs. As a result, the training loss of mask tokens is composed of teacher's task loss and mask diversity loss:

$$\mathcal{L}_{\text{token}} = \mathcal{L}_{\text{task}}^{(t)} + \mu \mathcal{L}_{\text{div}}, \tag{6}$$

where $\mu$ is a factor for balancing the loss. Note that we simply set $\mu = 1$ in all experiments as the mask tokens are easy to converge[2], and the masked feature $\hat{\boldsymbol{F}}^{(t)}$ is only used for learning semantic-aware mask tokens. In the distillation stage, we use the teacher's original feature to get predictions.

---

[2]For example, the teacher *Cascade Mask RCNN ResNeXt101* has 45.6 mAP on COCO val set, and the masked one is 45.4.

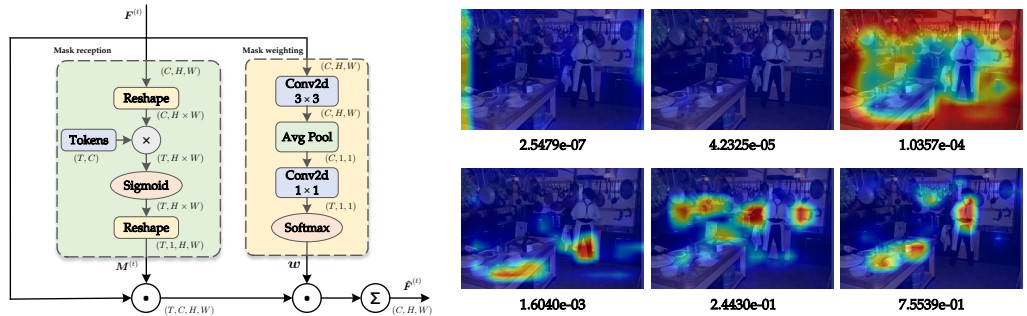

(a) **Computation of masked features.**   (b) **Visualization of learned masks and weights.**

Figure 3: (a) Illustration of the computation of masked features. We learn receptive tokens to generate the masks containing pixels of interests, then multiply the masks to the original features to get the masked features, along with a mask weighting module to learn the mask importance. (b) Visualization of learned masks and their corresponding importance on stage 4 of FPN in Faster RCNN-R101.

**Mask weighting module.** By learning mask tokens, we can obtain multiple masks of different PoIs, yet not all masks are of the same importance. For example, we would assign larger loss weights on foreground distillations than on background, as the foreground regions are usually more important in detection. In this paper, we thus propose a mask weighting module to determine the mask importance independently for each image. Concretely, we conduct a simple convolution-based module (see Figure 3 (a)) to predict the weights of each mask in each feature map, then weight the importance vector $w$ onto the masked features,

$$\hat{\boldsymbol{F}}^{(t)} = \sum_{i=1}^{T} w_i (\boldsymbol{M}_i^{(t)} \odot \boldsymbol{F}^{(t)}). \tag{7}$$

Therefore, the masks on important pixels could obtain larger weights, while the pixels redundant to the task get small weights. As illustrated in Figure 3 (b), we visualize the masks and the corresponding importance on the last stage (stride $= 32$) of FPN. The background and meaningless empty masks have negligible weights, while the masks associated with real objects have larger weights.

## 4.2 DISTILLATION WITH LEARNED MASKS

Now we formulate our feature distillation in the knowledge distillation stage. With the learned masks $\boldsymbol{M}^{(t)}$, a straightforward way is to reconstruct the teacher's feature is adopting Eq.(2) as previous methods. However, as previously discussed in Section 4.1, our masks contain both important PoIs and less valuable ones, and we thus propose a mask weighting module to capture the importance of each mask, which can be used in distillation for better balance.

**Instance-dependent mask weights.** With the learned mask importance in Section 4.1, we can naturally adopt these importance weights as the loss weights on each mask; *i.e.*, if a mask is critical to the performance, it should be highlighted in teacher-student knowledge transfer. Therefore, Eq.(2) can be reformulated as

$$\mathcal{L}_{\text{feature}} = \sum_{i=1}^{T} \frac{w_i}{C \sum_{j=1}^{H \times W} M_{i,j}} \left\| \boldsymbol{M}_i^{(t)} \odot \boldsymbol{F}^{(t)} - \boldsymbol{M}_i^{(t)} \odot \phi(\boldsymbol{F}^{(s)}) \right\|_2^2. \tag{8}$$

Note that the mask importance is determined instance-wisely, and thus our adaptive mask loss weights can be more precise and effective than pre-assigning fixed loss weights in a heuristic way.

**Customizing masks with student feature.** In feature distillation, it is difficult for the student to reconstruct all the feature maps from the teacher finely due to their capacity gap. Forcing the student to learn from the teacher would disturb the optimization for those hard-to-reconstruct pixels. Besides, some pixels that are meaningless to the student could also exist, and the unnecessary reconstructions on them may also weaken the distillation performance. In MasKD, we propose to alleviate these problems by refining the masks with the student feature.

The teacher's learned mask tokens can provide a precise observation of the reconstruction rate of the student's feature, *i.e.*, if a pixel fully recovers the teacher's corresponding feature, their similarities to the mask tokens will be the same. In contrast, the similarities will be different when the features vary significantly. Therefore, we propose to customize the mask regions by multiplying the student's mask regions, and the distillation pixels should consist of the ones both important to the teacher and student. In contrast, the noise pixels in the teacher's feature map will be ignored. Our eventual feature distillation loss can be formulated as

$$\mathcal{L}_{\text{MasKD}} = \sum_{i=1}^{T} \frac{w_i}{C \sum_{j=1}^{H \times W} M_{i,j}^{(r)}} \left\| M_i^{(r)} \odot F^{(t)} - M_i^{(r)} \odot \phi(F^{(s)}) \right\|_2^2, \tag{9}$$

where $M^{(r)} = M^{(s)} \odot M^{(t)}$, and $M^{(s)} = \sigma(E\phi(F^{(s)}))$ is generated by the teacher's learned mask tokens $E$ and student's feature $F^{(s)}$. Notably, to avoid masking out critical regions by the student's mask, we add a warmup stage in the early training period, and perform mask customization after the warmup to ensure a sufficient convergence of student.

**Overall loss function.** Following the previous detection KD method GID (Dai et al., 2021), we also adopt a distillation on the predicted bounding box regressions. Therefore, as the overall knowledge distillation framework illustrated in Figure 2, the overall loss function of student is formulated as

$$\mathcal{L}_{\text{student}} = \mathcal{L}_{\text{task}} + \lambda_1 \mathcal{L}_{\text{MasKD}} + \lambda_2 \mathcal{L}_{\text{reg-kd}}(r_t, r_s), \tag{10}$$

where $r_t$ and $r_s$ are regression predictions of teacher and student, $\lambda_1$ and $\lambda_2$ are factors for balancing the losses. Note that we do not adopt distillation on the classification outputs, as the classification losses vary in different detectors (*e.g.*, SoftMax CrossEntropy loss in Faster R-CNN (Ren et al., 2015) and Sigmoid Focal loss in RetinaNet (Lin et al., 2017b)), we empirically find that their suitable loss weights on classification KD are quite different, requiring a large amount of resources in hyper-parameter tuning compared to $\lambda_1$ and $\lambda_2$. As a result, we remove the classification distillation for better generalizability and transferability of our method, though it could gain further improvements.

## 5 EXPERIMENTS

In this section, to show our superiority and generalizability, we conduct experiments on two popular dense prediction tasks: object detection and semantic segmentation. Furthermore, we also experiment on image classification task to show that our MasKD can also achieve improvements on it.

### 5.1 OBJECT DETECTION

We first validate our efficacy on object detection task. We conduct experiments on MS COCO dataset (Lin et al., 2014) following previous KD works (Wang et al., 2019; Dai et al., 2021; Du et al., 2021), and evaluate the networks with average precision (AP) on COCO *val2017* set.

**Network architectures.** For comprehensive experiments, we first follow (Wang et al., 2019; Dai et al., 2021; Du et al., 2021), and adopt multiple detection frameworks for our baseline settings, including two-stage detector Faster-RCNN (Ren et al., 2015), one-stage detector RetinaNet (Lin et al., 2017b), and anchor-free detector FCOS (Tian et al., 2019b). We take ResNet-101 (R101) (He et al., 2016) backbone as the teacher network, with ResNet-50 (R50) as the student. Follow (Zhang & Ma, 2020; Yang et al., 2021) methods, we conduct experiments on stronger teacher detectors, including two-stage detector Cascade Mask RCNN (Cai & Vasconcelos, 2018), one-stage detector RetinaNet (Lin et al., 2017b), and anchor-free detector RepPoints (Yang et al., 2019), with stronger backbone ResNeXt101 (X101) (Xie et al., 2017).

**Training strategies.** We train our mask tokens for 2000 iterations using an Adam optimizer with $0.001$ weight decay, and a cosine learning rate decay is adopted with an initial value of $0.01$. In the distillation stage, we follow the standard $2\times$ schedule on detection as previous works (Wang et al., 2019; Dai et al., 2021; Du et al., 2021; Yang et al., 2021). For the loss weights, we simply set $\lambda_1$ and $\lambda_2$ to 1 in Eq.(10) on Faster RCNN-R50 student, and the weights on other architecture variants are adjusted to keep a similar amount of loss values as Faster RCNN-R50. Detailed training strategies can be found in Appendix A.5.

Table 1: **Object detection performance with baseline settings on COCO val set.** T: teacher. S: student. †: inheriting strategy (Yang et al., 2021) adopted. References for the methods can be found in Appendix A.1.

| Method | AP | AP$_{50}$ | AP$_{75}$ | AP$_S$ | AP$_M$ | AP$_L$ |
|---|---|---|---|---|---|---|
| *Two-stage detectors* | | | | | | |
| T: Faster RCNN-R101 | 39.8 | 60.1 | 43.3 | 22.5 | 43.6 | 52.8 |
| S: Faster RCNN-R50 | 38.4 | 59.0 | 42.0 | 21.5 | 42.1 | 50.3 |
| Fitnet | 38.9 | 59.5 | 42.4 | 21.9 | 42.2 | 51.6 |
| GID | 40.2 | 60.7 | 43.8 | 22.7 | 44.0 | 53.2 |
| FRS | 39.5 | 60.1 | 43.3 | 22.3 | 43.6 | 51.7 |
| FGD | 40.4 | - | - | 22.8 | 44.5 | 53.5 |
| MasKD | 40.8 | 60.7 | 44.4 | 23.2 | 44.6 | 53.6 |
| MasKD† | **41.0** | **60.8** | **44.6** | **23.5** | **45.0** | **53.9** |
| *One-stage detectors* | | | | | | |
| T: RetinaNet-R101 | 38.9 | 58.0 | 41.5 | 21.0 | 42.8 | 52.4 |
| S: RetinaNet-R50 | 37.4 | 56.7 | 39.6 | 20.0 | 40.7 | 49.7 |
| Fitnet | 37.4 | 57.1 | 40.0 | 20.8 | 40.8 | 50.9 |
| GID | 39.1 | 59.0 | 42.3 | 22.8 | 43.1 | 52.3 |
| FRS | 39.3 | 58.8 | 42.0 | 21.5 | 43.3 | 52.6 |
| FGD | 39.6 | - | - | 22.9 | 43.7 | 53.6 |
| MasKD | 39.8 | 59.0 | 42.5 | 21.5 | 43.9 | 54.0 |
| MasKD† | **39.9** | **59.0** | **42.5** | **23.3** | **43.9** | **54.4** |
| *Anchor-free detectors* | | | | | | |
| T: FCOS-R101 | 40.8 | 60.0 | 44.0 | 24.2 | 44.3 | 52.4 |
| S: FCOS-R50 | 38.5 | 57.7 | 41.0 | 21.9 | 42.8 | 48.6 |
| Fitnet | 39.9 | 58.6 | 43.1 | 23.1 | 43.4 | 52.2 |
| GID | 42.0 | 60.4 | 45.5 | 25.6 | 45.8 | 54.2 |
| FRS | 40.9 | 60.3 | 43.6 | 25.7 | 45.2 | 51.2 |
| FGD | 42.1 | - | - | 27.0 | 46.0 | 54.6 |
| MasKD | 42.6 | 61.2 | 46.3 | 26.5 | **46.9** | 54.2 |
| MasKD† | **42.9** | **61.5** | **46.5** | **27.2** | 46.8 | **54.9** |

Table 2: **Object detection performance with stronger teachers on COCO val set.** *CM RCNN*: Cascade Mask RCNN. †: inheriting strategy adopted. References for the methods can be found in Appendix A.1.

| Method | AP | AP$_{50}$ | AP$_{75}$ | AP$_S$ | AP$_M$ | AP$_L$ |
|---|---|---|---|---|---|---|
| *Two-stage detectors* | | | | | | |
| T: CM RCNN-X101 | 45.6 | 64.1 | 49.7 | 26.2 | 49.6 | 60.0 |
| S: Faster RCNN-R50 | 38.4 | 59.0 | 42.0 | 21.5 | 42.1 | 50.3 |
| LED | 38.7 | 59.0 | 42.1 | 22.0 | 41.9 | 51.0 |
| FGFI | 39.1 | 59.8 | 42.8 | 22.2 | 42.9 | 51.1 |
| COFD | 38.9 | 60.1 | 42.6 | 21.8 | 42.7 | 50.7 |
| FKD | 41.5 | 62.2 | 45.1 | 23.5 | 45.0 | 55.3 |
| FGD | 42.0 | - | - | 23.7 | 46.4 | 55.5 |
| MasKD | 42.4 | 62.9 | 46.8 | 24.2 | 46.7 | 55.9 |
| MasKD† | **42.7** | **63.1** | **47.0** | **24.5** | **47.4** | **56.2** |
| *One-stage detectors* | | | | | | |
| T: RetinaNet-X101 | 41.2 | 62.1 | 45.1 | 24.0 | 45.5 | 53.5 |
| S: RetinaNet-R50 | 37.4 | 56.7 | 39.6 | 20.0 | 40.7 | 49.7 |
| COFD | 37.8 | 58.3 | 41.1 | 21.6 | 41.2 | 48.3 |
| FKD | 39.6 | 58.8 | 42.1 | 22.7 | 43.3 | 52.5 |
| FRS | 40.1 | 59.5 | 42.5 | 21.9 | 43.7 | 54.3 |
| FGD | 40.4 | - | - | 23.4 | 44.7 | 54.1 |
| MasKD | 40.9 | 60.1 | 43.6 | **22.8** | 45.3 | 55.1 |
| MasKD† | **41.0** | **60.2** | **43.8** | 22.6 | **45.3** | **55.3** |
| *Anchor-free detectors* | | | | | | |
| T: RepPoints-X101 | 44.2 | 65.5 | 47.8 | 26.2 | 48.4 | 58.5 |
| S: RepPoints-R50 | 38.6 | 59.6 | 41.6 | 22.5 | 42.2 | 50.4 |
| FKD | 40.6 | 61.7 | 43.8 | 23.4 | 44.6 | 53.0 |
| FGD | 41.3 | - | - | 24.5 | 45.2 | 54.0 |
| MasKD | 41.8 | 62.6 | 45.1 | 24.2 | 45.4 | 55.2 |
| MasKD† | **42.5** | **63.4** | **45.8** | **24.9** | **46.1** | **56.8** |

**Distillation settings.** We conduct feature distillation on the predicted feature maps, and train the student with our MasKD loss $\mathcal{L}_{\text{MasKD}}$, regression KD loss, and task loss, as formulated in Eq.(10). The number of tokens is set to 6 in all experiments.

**Experimental results.** Our results compared with previous methods are summarized in Table 1 and Table 2. In Table 1, we first compare KD methods with baseline settings, where the teacher and student are the same ResNet (He et al., 2016) variants. Our MasKD can significantly surpass the other state-of-the-art methods. For example, the RetinaNet with ResNet-50 backbone gets 2.5 AP improvement on COCO. We further investigate our efficacy on stronger teachers whose backbones are replaced by stronger ResNeXts. As in Table 2, student detectors achieve more enhancement on both AP and AR with our MasKD, especially when with a Cascade Mask RCNN-X101 teacher, MasKD gains a significant improvement of 3.8 AP over the Faster RCNN-R50.

Our MasKD outperforms existing state-of-the-art methods consistently on all the popular model settings. Note that our method is more general to dense prediction tasks without any ground-truth prior or manual heuristics on specific detection frameworks, showing that our mask tokens effectively locate the important PoIs that benefit the distillation.

## 5.2 SEMANTIC SEGMENTATION

Learning masks in semantic segmentation task is straightforward as it needs to predict fine-grained pixel-wise semantic classifications. We conduct experiments on Cityscapes dataset(Cordts et al.,

Table 3: **Semantic segmentation results on Cityscapes dataset.** †: trained from scratch. Other models are pretrained on ImageNet. FLOPs is measured based on an input size of $1024 \times 2048$. References for the methods can be found in Appendix A.1.

| Method | Params (M) | FLOPs (G) | mIoU (%) Val | Test | Method | Params (M) | FLOPs (G) | mIoU (%) Val | Test |
|---|---|---|---|---|---|---|---|---|---|
| T: DeepLabV3-R101 | 61.1 | 2371.7 | 78.07 | 77.46 | T: DeepLabV3-R101 | 61.1 | 2371.7 | 78.07 | 77.46 |
| S: DeepLabV3-R18 | | | 74.21 | 73.45 | S: DeepLabV3-MBV2 | | | 73.12 | 72.36 |
| SKD | | | 75.42 | 74.06 | SKD | | | 73.82 | 73.02 |
| IFVD | 13.6 | 572.0 | 75.59 | 74.26 | IFVD | 3.2 | 128.9 | 73.50 | 72.58 |
| CWD | | | 75.55 | 74.07 | CWD | | | 74.66 | 73.25 |
| CIRKD | | | 76.38 | 75.05 | CIRKD | | | **75.42** | 74.03 |
| MasKD | | | **77.00** | **75.59** | MasKD | | | 75.26 | **74.23** |
| S: DeepLabV3-R18† | | | 65.17 | 65.47 | S: PSPNet-R18 | | | 72.55 | 72.29 |
| SKD | | | 67.08 | 66.71 | SKD | | | 73.29 | 72.95 |
| IFVD | 13.6 | 572.0 | 65.96 | 65.78 | IFVD | 12.9 | 507.4 | 73.71 | 72.83 |
| CWD | | | 67.74 | 67.35 | CWD | | | 74.36 | 73.57 |
| CIRKD | | | 68.18 | 68.22 | CIRKD | | | 74.73 | 74.05 |
| MasKD | | | **73.95** | **73.74** | MasKD | | | **75.34** | **74.61** |

2016) following previous KD methods (Wang et al., 2020; Shu et al., 2021; Yang et al., 2022), and evaluate the networks with mean Intersection-over-Union (mIoU) on Cityscapes *val* and *test* sets.

**Network architectures.** For all experiments, we use DeepLabV3 (Chen et al., 2018) framework with ResNet-101 (R101) (He et al., 2016) backbone as the teacher network. While for the students, we use various frameworks (DeepLabV3 and PSPNet (Zhao et al., 2017)) and backbones (ResNet-18 and MobileNetV2 (Sandler et al., 2018)) to valid the effectiveness of our method.

**Training strategies.** Following CIRKD (Yang et al., 2022), we adopt a standard data augmentation, which consists of random flipping, random scaling in the range of $[0.5, 2]$, and a crop size of $512 \times 1024$. We train the models using an SGD optimizer with a momentum of $0.9$, and a polynomial annealing learning rate scheduler is adopted with an initial value of $0.02$. We train the mask tokens for 2000 iterations in the mask learning stage, and then train the student for 40000 iterations.

**Distillation settings.** We conduct feature distillation on the predicted segmentation maps, and train the student with our MasKD loss $\mathcal{L}_{\text{MasKD}}$ and task loss. Note that we use the distillation loss in Eq.(2) and do not involve mask weighting and mask customization as in detection, since all the regions in segmentation are equally important. The loss weight of $\mathcal{L}_{\text{MasKD}}$ is set to $0.5$, and the number of tokens is set to $8$ for all experiments.

**Experimental results.** Our results compared with previous methods are summarized in Table 3. By simply distilling different mask regions separately, our MasKD significantly outperforms state-of-the-art methods, especially when the student is randomly initialized without pretraining on ImageNet. For example, MasKD achieves 73.74% mIoU on Cityscapes test set with a trained-from-scratch DeepLabV3-R18 student, while the previous state-of-the-art method CIRKD obtains 68.22%. This indicates that the semantic regions can help students learn better on semantic segmentation tasks. Besides, we also visualize the learned masks on Cityscapes in Appendix A.7, which shows that 8 mask tokens can generate promising segmentation results compared to the ground-truth labels.

## 5.3 ABLATION STUDY

**Effects of components in MasKD.** We perform experiments to show the effects of each proposed component in MasKD in Table 4. **Feature distillation with random regions.** Compared to the mimic baseline, distillation with randomly-initialized mask tokens can also gain improvements, as it still captures weak segmentations on the feature (see Figure 4 (a)). **+ mask divergence loss.** As shown in Figure 4 (b), the masks learned without task loss can also obtain fairly good segmentations, but lack task knowledge (*e.g.*, it mixtures the background and foreground pixels into one mask). The

result is even 0.4 worse than the random tokens, indicating that the task-related masks are important in MasKD. **+ task loss.** Using task loss can learn better meaningful masks to the task, and thus gains a further improvement of 0.6 compared to the random tokens. **+ adaptive mask loss weights.** Adaptive mask loss weights provide a better balance on the distillation regions, thus having an additional 0.2 gain. **+ mask customization.** We adopt the student's feature to customize the masks generated with the teacher's feature, and the results show that it can obtain a further 0.2 increment. Our MasKD with all components can achieve a significant 1.5 (41.4 vs. 39.9) improvement over the mimic baseline.

Table 4: **Ablation of components in MasKD.** We train the student *Faster RCNN-R50* with teacher *Cascade Mask RCNN-X101* using $1\times$ schedule, and only adopts distillation on feature. The AP of mimic baseline is 39.9.

| Mask learning | | Feature distillation | | AP |
|---|---|---|---|---|
| $\mathcal{L}_{\text{div}}$ | $\mathcal{L}_{\text{task}}^{(t)}$ | mask weighting | mask custom. | |
| ✗ | ✗ | ✗ | ✗ | 40.4 |
| ✓ | ✗ | ✗ | ✗ | 40.0 |
| ✗ | ✓ | ✗ | ✗ | 40.6 |
| ✓ | ✓ | ✗ | ✗ | 41.0 |
| ✓ | ✓ | ✓ | ✗ | 41.2 |
| ✓ | ✓ | ✓ | ✓ | **41.4** |

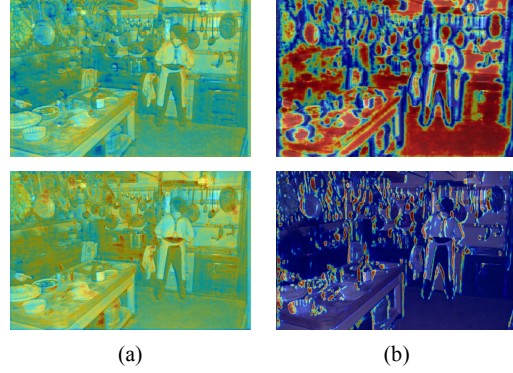

(a)          (b)

Figure 4: (a) Masks with randomly-initialized tokens. (b) Masks with tokens learned with only $\mathcal{L}_{\text{div}}$.

**Using ground-truth semantic masks on semantic segmentation.** The semantic segmentation task provides accurate fine-grained segmentations in ground-truth annotations, which can be regarded as the masks in our masked distillation. We conduct experiments to compare the ground-truth semantic masks with our learned masks. Concretely, with 19 classes and 1 ignore class in Cityscapes dataset, we generate 20 masks, where the points with the corresponding class in the mask are set to 1, and others are set to 0. We train the student with these generated masks and our learned masks separately using Eq.(2). As shown in Table 5, using ground-truth masks can also obtain a very high performance, which outper-

Table 5: **Effect of using ground-truth masks in semantic segmentation.** We use DeepLabV3-R101 as teacher network, and train students with R18 backbone, then report the evaluation results on val set.

| Mask | DeepLabV3 | PSPNet |
|---|---|---|
| gt. | 76.71 | 75.26 |
| learned | **77.00** | **75.34** |

forms existing KD methods, yet our MasKD achieves even higher performance. One possible reason is that our learned masks contain soft probabilities on each point and thus have more information to teach the student.

## 6   CONCLUSION

Feature distillation matters for dense prediction tasks as the feature contains both recognition and localization information. In this paper, unlike previous object detection methods that usually perform masked distillation on bounding boxes, we present a new mask generation method by locating the pixels of interests using learnable receptive tokens. Our method enjoys finer pixel-level feature reconstruction and better generalizability without bounding box priors. Besides, our additional adaptations of receptive tokens in distillation loss improve the performance by making the student focus more on those important pixels. Extensive experiments on object detection and semantic segmentation tasks validate our efficacy.

**Acknowledgements.** This work was supported in part by the Australian Research Council under Project DP210101859 and the University of Sydney Research Accelerator (SOAR) Prize.

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

## A APPENDIX

### A.1 REFERENCES OF COMPARED KD METHODS

**Object detection.** Fitnet (Romero et al., 2014), GID (Dai et al., 2021), FRS (Du et al., 2021), FGD (Yang et al., 2021), LED (Chen et al., 2017), FGFI (Wang et al., 2019), COFD (Heo et al., 2019), FKD (Zhang & Ma, 2020).

**Semantic segmentation.** SKD (Liu et al., 2019), IFVD (Wang et al., 2020), CWD (Shu et al., 2021), CIRKD (Yang et al., 2022).

### A.2 ARCHITECTURE OF MASK WEIGHTING MODULE

As shown in Figure 3 (a), the mask weighting module (yellow box) first takes distillation feature $F^{(t)}$ as input, then conducts a sequential of $3 \times 3$ convolution and average pooling to get the feature of the whole image, then a $1 \times 1$ convolution is adopted to predict the importance scores of $T$ mask tokens.

### A.3 DISTILLATION RESULTS ON IMAGE CLASSIFICATION

In previous experiments, we show that MasKD has obvious effects on dense prediction tasks (object detection and semantic segmentation). As our method could be applied to arbitrary tasks, we further conduct experiments to validate our efficacy on image classification task.

Following CRD (Tian et al., 2019a) and DIST (Huang et al., 2022c), we train ResNet-18 with ResNet-34 teacher on ImageNet, and adopt distillations with our MasKD and the baseline mimic, respectively. We perform feature distillations on the outputs of the last four stages of the networks. As the results shown in Table 6, mimicking the backbone features could obtain similar performance compared to the KD (Hinton et al., 2015) baseline, while our MasKD can improve mimic by $0.55\%$, showing that our masked distillation can also benefit the feature distillation on classification task.

Table 6: **Classification performance on ImageNet.** Teacher: ResNet-34. Student: ResNet-18.

| Teacher | Student | KD (Hinton et al., 2015) | CRD Tian et al. (2019a) | DIST Huang et al. (2022c) | Mimic | MasKD |
|---------|---------|--------------------------|-------------------------|---------------------------|-------|-------|
| 73.31 | 69.76 | 70.66 | 71.17 | 72.07 | 70.71 | 71.26 |

## A.4 MORE ABLATION STUDIES

**Ablation on the number of tokens.** We conduct experiments on COCO dataset to investigate the effects of different numbers of mask tokens in MasKD. As shown in Table 7, with only 2 tokens, MasKD improves mimic by $1.4\%$ AP, while more tokens could achieve further improvements. In this paper, we choose a moderate number of 6 for a better performance-efficiency trade-off.

Table 7: **Ablation on the number of tokens in MasKD.** We train the student *Faster RCNN-R50* with teacher *Mask RCNN-X101* using $1\times$ schedule.

| 0 (mimic) | 2 | 6 | 10 | 14 |
|-----------|-----|-----|-----|-----|
| 39.9 | 41.3 | 41.4 | 41.4 | 41.5 |

## A.5 DETAILED TRAINING SETTINGS ON OBJECT DETECTION

We conduct experiments on object detection using COCO dataset (Lin et al., 2014), and evaluate our performance on COCO *val2017* set. In Table 1 and Table 2, we report our main results on various detection frameworks such as *Faster RCNN (Ren et al., 2015)*, *RetinaNet (Lin et al., 2017b)*, *FCOS (Tian et al., 2019b)*, and *RepPoints (Yang et al., 2019)*. All the models are trained with the official strategies of $2\times$ schedule in MMDetection (Chen et al., 2019).

**Loss weights**: We set MasKD loss weight $\lambda_1 = 1$ and regression loss weight $\lambda_2 = 1$ on *Faster RCNN* students. For other detection frameworks, we simply adjust the loss weight $\lambda_1$ of $\mathcal{L}_{\text{MasKD}}$ to keep a similar amount of loss value as *Faster RCNN*. Concretely, the loss weights $\lambda_1$ on *RetinaNet*, *FCOS*, and *RepPoints* are 5, 10, and 10, respectively.

## A.6 VISUALIZATION OF LEARNED MASKS ON COCO

We visualize the learned complete masks of *Faster RCNN-R101* teacher in Figure 5. We can see that (1) The earlier stages in FPN prefers to recognize small objects, while the stage 4 is used to detect large objects. (2) The effective region for representing an object is usually smaller than the object, which indicates that we may not need to reconstruct all the pixels in the bounding box in distillation, and many of them are useless background information. (3) The mask tokens are associated with different objects, while one mask token is learned to cover all the remaining background pixels, which shows that the background features are also helpful for detection. (4) Some objects that are not marked in the ground-truth annotations can also be included in our masks, and these foreground features are also informative and useful to the student, this could be a superiority of our MasKD compared to previous methods with ground-truth priors.

## A.7 VISUALIZATION OF LEARNED MASKS ON CITYSCAPES

We visualize the learned masks of DeepLabV3-R101 teacher on Cityscapes dataset in Figure 6. We can see that, the learned masks on semantic segmentation task is much clear than those on object detection, and the edges of objects can be precisely masked. This is because the semantic segmentation task provides a direct supervision on the segmentations, thus the features are more discriminative on various semantic types.

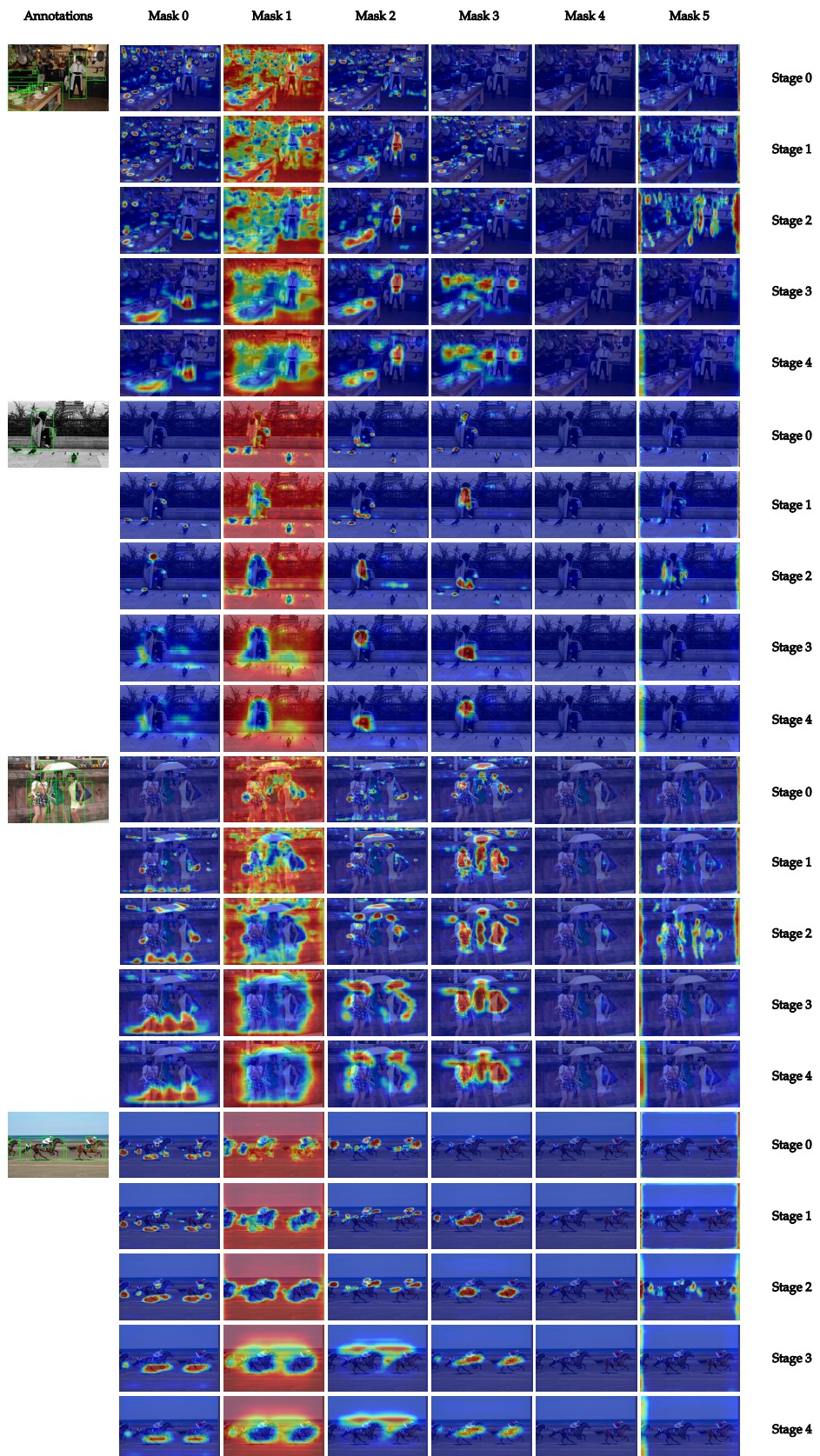

Figure 5: **Complete visualization of learned masks on COCO dataset.** Zoom up to view better.

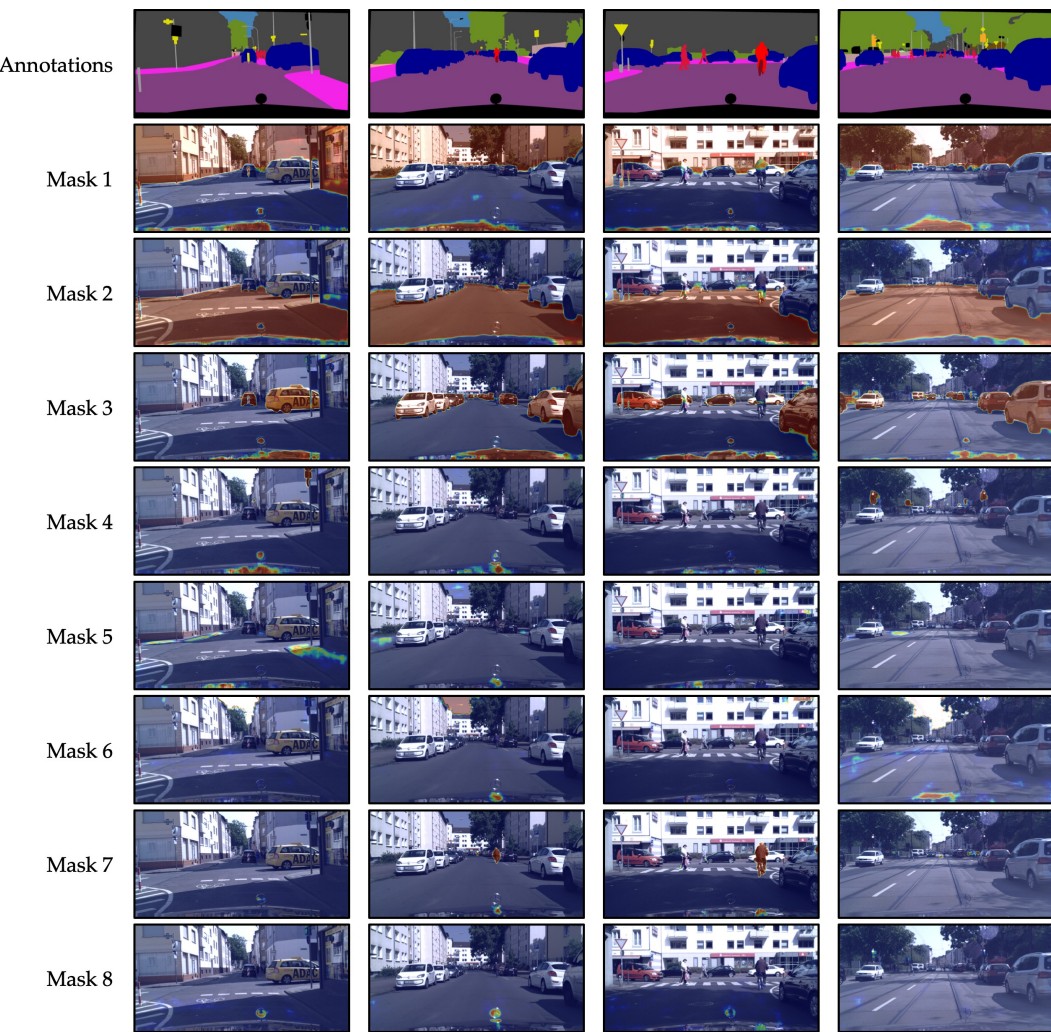

Figure 6: **Complete visualization of learned masks on Cityscapes dataset.** Zoom up to view better.

