# OpenReview forum: "Masked Distillation with Receptive Tokens"
_ICLR.cc/2023/Conference — ICLR 2023 poster_

### Official Review · Reviewer_EXB7 · 2022-10-23

**Confidence:** 4
**Correctness:** 3
**Technical Novelty And Significance:** 2
**Empirical Novelty And Significance:** 2
**Recommendation:** 5

**Clarity, Quality, Novelty And Reproducibility:**

- Overall, the paper is well written. The background and the proposed method are well introduced. Figures are also okay.
- There is no mask weighting or refinement for semantic segmentation. This only becomes clear late in the paper. I think this distinction should be made clear earlier in the paper, when these methods are introduced.
- How are the weights in Eq. 7 learned? Is it the same as the masks, with the task loss on a frozen teacher model? This is unclear in the paper.
- Footnote 2 on page 4 does not say anything about "easy to converge", only that at convergence the original performance is almost fully recovered. The text says that mask tokens are easy and quick to train, but this is not apparent from the footnote.
- The first paragraph on page 6 is written in a complicated way. It is assumed that a mask for the student can also be created, but this is not obvious. I would first state that the learned embeddings are used to create a student mask.
- Table 4: What's the difference between the gray and the black marks (crosses and ticks)?
- Figure 1: What activations are visualized here? Are these CAM activations?

**Strength And Weaknesses:**

Strengths:
- The proposed method for generating masks is task-agnostic in a sense that no priors about the output space (bounding boxes or segmentation maps) are used.
- It is good to see multiple detection and segmentation architectures and backbones evaluated.
- The method is simple and appears to be consistently effective across model designs

Weaknesses:
- I found the motivation for identifying (un)important regions/pixels in a task-agnostic manner unsatisfactory. Most of the motivation builds on foreground vs. background objects. But this is only valid for object detection, not so much for semantic segmentation. The better motivation seems to be a "balanced loss/impact" for different regions when doing distillation.
- I'm do not fully understand why it is "optimal" that only the pixels where both teacher and student masks agree should be distilled. This seems to be a guess (or a random justification) for something that works practically (i.e., the experiments show improvements).
- Overall, I found the experimental setting to be insufficient.
  - I'm missing more recent detectors like CenterNet [A] or DETR-based models like DINO [B-D].
  - Choosing R101 and R50 as teacher and student, respectively, seems odd to me. The performance difference is not big between these two backbones (Table 1). Isn't the more practical situation for model distillation to have a very strong model distilling information into a much smaller one? I see such experiments for segmentation (with MobileNet backbones), but shouldn't this be the default for most experiments?
  - Evaluations are only done on two benchmarks (COCO and CityScapes), although the method is advertised as general and task-agnostic. Given the rather small improvements over prior works across all experiments, I think additional evaluations on other datasets are needed to confirm consistent improvements of the proposed method over existing ones.
- Significance of the improvement over prior works
  - The improvement over FGD is consistent but minor in all results of Tables 1 and 2. This relates to my comment above about the experimental setting. With such consistent but only minor improvements, I find it necessary to provide more experimental evidence in the form of (a) additional datasets, and (b) reporting mean errors with standard deviation over multiple runs with different random seeds.
  - I see similarly small margins in Table 4, where the proposed method outperforms randomly initialized tokens by only 0.6% AP. Again, are these improvements statistically significant?
  - Why is the improvement much bigger for models trained from scratch (w/o ImageNet pre-training)? Is there any intuition for that? Also, are the teacher models also trained from scratch? If no, is this a practical situation where the teacher model can use ImageNet-pretrained weights, but the student cannot?
- Table 1: Is there an explanation why the student model outperforms the teacher? The motivation and intuition of model distillation is to mimic the teacher's behavior. How does the model become better than the student?

References:
- [A] Objects as Points. Zhou et al. arXiv 1904.07850
- [B] DINO: DETR with Improved DeNoising Anchor Boxes for End-to-End Object Detection. Zhang et al. arXiv 2203.03605
- [C] Dn-detr: Accelerate detr training by introducing query denoising. Feng et al. CVPR 2022
- [D] DAB-DETR: Dynamic Anchor Boxes are Better Queries for DETR. Liu et al. ICLR 2022

**Summary Of The Paper:**

This paper is about model distillation for dense prediction tasks, specifically for object detection and semantic segmentation. Prior works on this topic identify per-region distillation as an important ingredient to a successful knowledge transfer. This paper proposes a novel way to identify what regions should be distilled and with what importance (weight). These regions are modeled as masks, can be trained based on the respective task loss, and are not biased by the form of ground truth used (bounding boxes or segmentation maps). Given a teacher network, embedding vectors (one per region, the number of regions being a hyper-parameter) are learned that create a per-pixel similarity map with the feature maps, and then a (soft) masking. These embeddings (along with a corresponding weighting) are learned such that (a) the sum of all masked features can minimize the task loss, and (b) the masks are diverse. In a second phase, these masks are used for knowledge distillation into a student model. Additionally, the student also builds a mask (from the learned embeddings and the feature maps) and the distillation then becomes dependent on the agreement between teacher and student masks. Experiments on both object detection and semantic segmentation demonstrate consistent improvement over existing methods on multiple base models and backbones.

**Summary Of The Review:**

In summary, I think the experimental setting needs to improve to justify and solidify the improvement over prior works. I lean towards rejection of the submission.

---

> ### Author Response · Authors · 2022-11-18
> **Response to Reviewer EXB7 (part I)**
>
> Dear Reviewer EXB7,
>
> Thank you for your efforts in reviewing our paper. We formulate the responses to your comments and questions as follows.
>
> **Q1: The better motivation seems to be a "balanced loss/impact" for different regions when doing distillation.**
>
> **A1:** We agree with your idea. In Introduction, for the convenience of understanding, we take the widely discussed foreground-background distillation in object detection as an example to formulate our motivation of learning and distilling different semantic regions independently. We will polish our writing to show our motivation more clearly.
>
> ---
>
> **Q2: I do not fully understand why it is "optimal" that only the pixels where both teacher and student masks agree should be distilled.**
>
> **A2:** We perform distillation on this intersection of masks after warmup; while in the warmup stage, the distillation masks are generated by the teacher only (i.e., all the teacher's preferences are desired to be transferred to the student). However, it is difficult for the student to fully reconstruct the teacher's feature because of the capacity gap, and some knowledge from the teacher might be meaningless to the student (e.g., the instance segmentation feature in mask R-CNN might be useless for a detection only student). Paying more attention on distilling these challenging and meaningless features would restrict the student from learning finer important features. As a result, we propose to relieve the burden of student after we know what regions are impossible to reconstruct through warmup.
>
> ---
>
> **Q3: The experimental setting is insufficient.**
>
> We need to clarify that for most knowledge distillation papers on object detection, the model settings in our Table 1 and Table 2 are acknowledged benchmark for comparing the performance. We also perform additional experiments according to your suggestions as follows.
>
> **Q3.1: More recent detectors like CenterNet and DETR-based models.**
>
> **A3.1:** Actually, we have conducted experiments on two recent anchor-free detectors similar to CenterNet, i.e., FCOS and RepPoints, and our improvements on those detectors are even more significant than the traditional detectors. For DETR-based models, we add experiments on Deformable DETR [1], and the results are summarized as follows.
>
> |Teacher|Student|Method|AP|AP$_{50}$|AP$_{75}$|AP$_S$|AP$_M$|AP$_L$|
> |:--:|:--:|:--:|:--:|:--:|:--:|:--:|:--:|:--:|
> |De-DETR-R101|-|-|46.8|66.3|50.7|30.3|49.8|61.5|
> |-|De-DETR-R18|-|43.7|62.3|47.5|26.2|45.7|59.0|
> |De-DETR-R101|De-DETR-R18|Mimic|43.9|62.6|47.2|26.1|46.6|58.8|
> |De-DETR-R101|De-DETR-R18|MasKD|44.3|63.2|47.6|26.5|47.1|59.7|
>
> Our MasKD also achieves obvious performance gains compared to student and mimic.
>
> **Q3.2: Having a very strong teacher distilling information to a much smaller one.**
>
> **A3.2:** Thanks for your suggestion. We take the widely-used light-weight detector RetinaNet-MobileNetV2 as the student, and use a heterogeneous and much stronger RetinaNet-R101 as the teacher. As the results summarized in the following table, our MasKD also outperforms the SOTA method GID on this setting.
>
> |Teacher|Student|Method|AP|AP$_{50}$|AP$_{75}$|AP$_S$|AP$_M$|AP$_L$|
> |:--:|:--:|:--:|:--:|:--:|:--:|:--:|:--:|:--:|
> |Retinanet-R101|-|-|38.9|58.0|41.5|21.0|42.8|52.4|
> |-|RetinaNet-Mbv2|-|31.0|48.9|32.7|16.4|33.8|42.6|
> |Retinanet-R101|RetinaNet-Mbv2|Mimic|33.0|50.0|35.0|15.6|35.7|47.6|
> |Retinanet-R101|RetinaNet-Mbv2|GID|33.5|51.9|35.5|19.2|36.9|44.3|
> |Retinanet-R101|RetinaNet-Mbv2|MasKD|33.8|51.0|35.7|16.5|36.4|48.1|
>
> **Q3.3: Additional evaluations on other datasets.**
>
> **A3.3:** We perform experiments on another typical detection dataset PASCAL VOC. We train Faster R-CNN and RetinaNet detectors with 1x scheduler (12 epochs), and the results comparing to mimic are as follows.
>
> |Teacher|Student|Method|mAP|AP$_{50}$|
> |:--:|:--:|:--:|:--:|:--:|
> |FasterRCNN-R101|-|-|52.0|80.8|
> |-|FasterRCNN-R50|-|49.5|79.4|
> |FasterRCNN-R101|FasterRCNN-R50|Mimic|51.5|79.8|
> |FasterRCNN-R101|FasterRCNN-R50|MasKD|52.3|80.2|
> |RetinaNet-R101|-|-|56.5|81.3|
> |-|RetinaNet-R50|-|53.9|79.7|
> |RetinaNet-R101|RetinaNet-R50|Mimic|54.8|80.4|
> |RetinaNet-R101|RetinaNet-R50|MasKD|56.5|81.1|
>
> Our MasKD also obtains significant improvements on PASCAL VOC dataset compared to mimic.
>
> ---
> **References**
>
> [1] Zhu, X., Su, W., Lu, L., Li, B., Wang, X. and Dai, J., 2020, September. Deformable DETR: Deformable Transformers for End-to-End Object Detection. In International Conference on Learning Representations.

---

> ### Author Response · Authors · 2022-11-18
> **Response to Reviewer EXB7 (part II)**
>
> **Q4: Significance of the improvement over prior works.**
>
> **Q4.1: Provide more experimental evidence on (a) additional datasets and (b) reporting mean errors with standard deviation over multiple runs with different random seeds.**
>
> **A4.1:** Thanks a lot for this valuable suggestion. (a) We have added experiments on PASCAL VOC dataset (see **A3.3**). (b) We conduct the experiments in Table 1 five times independently. The results summarized in the following table show that the performance of our MasKD is stable.
>
> |Teacher|Student|1|2|3|4|5|Mean|Std.|
> |:--:|:--:|:--:|:--:|:--:|:--:|:--:|:--:|:--:|
> |FasterRCNN-R101|FasterRCNN-R101|40.8|40.8|40.7|40.8|40.8|40.8|0.04|
> |Retinanet-R101|RetinaNet-R50|40.0|40.1|40.1|40.1|40.1|40.1|0.04|
> |FCOS-R101|FCOS-R50|42.6|42.5|42.7|42.6|42.6|42.6|0.07|
>
> **Q4.2: Similarly small margins in Table 4.**
>
> **A4.2:** Thank you for your concerns about our performance on object detection. In our method, we simply kept the same configurations of our distillation loss in all settings. While we find that the optimal hyper-parameters vary in different detectors, and the recent state-of-the-art FGD adopts quite different configurations (e.g., loss weight) on each detector. Therefore, we further conduct experiments on all the students in Table 6 with better configurations following FGD. The results are summarized as follows.
>
> |Teacher|Student|Method|AP|AP$_{50}$|AP$_{75}$|AP$_S$|AP$_M$|AP$_L$|
> |:--:|:--:|:--:|:--:|:--:|:--:|:--:|:--:|:--:|
> |FasterRCNN-R101|FasterRCNN-R50|FGD|40.4|-|-|22.8|44.5|53.5|
> |FasterRCNN-R101|FasterRCNN-R50|MasKD|40.8 (**+0.4**)|60.9|44.5|23.2|44.2|54.1|
> |Retinanet-R101	|RetinaNet-R50	|FGD	|39.6	|-	|-	|22.9	|43.7	|53.6|
> |Retinanet-R101	|RetinaNet-R50	|MasKD	|40.1 (**+0.5**)	|59.3	|42.8	|21.7	|43.9	|54.0|
> |FCOS-R101	|FCOS-R50	|FGD	|42.1	|-	|-	|27.0	|46.0	|54.6|
> |FCOS-R101	|FCOS-R50	|MasKD	|42.6 (**+0.5**)	|61.2	|46.3	|26.5	|46.9	|54.2|
> |CM RCNN-X101	|FasterRCNN-R50	|FGD	|42.0	|-	|-	|23.8	|46.4	|55.5|
> |CM RCNN-X101	|FasterRCNN-R50	|MasKD	|42.6 (**+0.6**)	|62.9	|46.3	|23.6	|46.6	|56.3|
> |RetinaNet-X101	|RetinaNet-R50	|FGD	|40.7	|-	|-	|22.9	|45.0	|54.7|
> |RetinaNet-X101	|RetinaNet-R50	|MasKD	|41.4 (**+0.7**)	|60.6	|44.1	|22.8	|45.7	|56.2|
> |Reppoints-X101	|Reppoints-R50	|FGD	|41.3	|-	|-	|24.5	|45.2	|54.0|
> |Reppoints-X101	|Reppoints-R50	|MasKD	|41.7 (**+0.4**)	|62.6	|45.0	|24.1	|45.8	|55.0|
>
> We believe the improvements are significant enough to show our superiority. Moreover, FGD is designed for object detection only (it leverages the ground-truth bounding boxes to compute the loss), while our MasKD is more generic and can achieve state-of-the-art performance consistently on object detection and semantic segmentation tasks.
>
> **Q4.3: Why is the improvement much bigger for models trained from scratch (w/o ImageNet pre-training)? Is there any intuition for that? Also, are the teacher models also trained from scratch? If no, is this a practical situation where the teacher model can use ImageNet-pretrained weights, but the student cannot?**
>
> **A4.3:**
>
> 1. Without ImageNet pre-training, the randomly-initialized model is more difficult to perform well using the limited training data. In contrast, our MasKD provides better supervision information by using the teacher's feature with is pixel-wise region-based distillation.
> 2. For fair comparisons, we follow the same settings in CWD and CIRKD (specifically, we use the code released by CIRKD to implement our MasKD), where the teacher is also trained from scratch.

---

> ### Author Response · Authors · 2022-11-18
> **Response to Reviewer EXB7 (part III)**
>
> **Q5: Table 1: Is there an explanation why the student model outperforms the teacher? The motivation and intuition of model distillation is to mimic the teacher's behavior. How does the model become better than the student?**
>
> **A5:** This situation is common as students in GID and FGD also outperforms the teachers. In Table 1, the models are trained with 2x schedule (24 epochs), and it is acknowledged that 3x schedule can improve the performance further. We think the reason of "student surpasses teacher" is that, the teacher models are not fully converged on this training schedule; while for student, the supervision from teacher can make it converge faster, and thus outperforms the teacher. For example, in [mmdetection](https://github.com/open-mmlab/mmdetection/tree/master/configs/retinanet), RetinaNet-R101 can be further improved from 38.9 (2x schedule) to 41.0 (3x schedule), while our distilled RetinaNet-R50 obtains 40.1 mAP.
>
> ---
>
> **Q6: How are the weights in Eq. 7 learned? Is it the same as the masks, with the task loss on a frozen teacher model?**
>
> **A6:** Yes, the $\mathbf{w}_i$ in Eq. (7) is optimized jointly with the mask tokens using the task loss on a frozen teacher model, as we want to use the task loss to identify the importance of each mask.
>
> ---
>
> **Q7: Table 4: What's the difference between the gray and the black marks (crosses and ticks)?**
>
> **A7:** To make the changes in each row more apparent, we use black and gray marks to represent changed settings and unchanged ones, respectively.
>
> ---
>
> **Q8: Figure 1: What activations are visualized here? Are these CAM activations?**
>
> **A8:** All the visualizations in this Figure 1, 3, 4, 5, and 6 are generated by our learned masks. Concretely, we adopt the learned mask tokens to generate their attentive regions using Eq. (3), then we overlay these probabilistic heatmaps onto the input image and get the visualizations, where the red color denotes high activation score and blue color denotes low activation score.
>
> ---
>
> The authors want to thank Reviewer EXB7 again for the detailed and valuable comments. We believe these comments and suggestions are important for polishing our work better. Hope our responses can address your concerns.

---

> ### Author Response · Authors · 2022-11-29
> **Further discussion to Reviewer EXB7**
>
> Dear Reviewer EXB7,
>
> We sincerely thank you for your efforts in reviewing our paper. We have provided corresponding responses and results, which we believe have covered your concerns. We hope to further discuss with you whether your concerns have been addresses or not. Please let us know if you still have any unclear part of our work.
>
> Best,
> Authors

---

> > ### Comment · Reviewer_EXB7 · 2022-11-29
> > **Re: Further discussion to Reviewer EXB7**
> >
> > Dear authors,
> >
> > I very much appreciate your efforts in the rebuttal phase, especially the additional explanations and experiments. Most of my comments and questions have been addressed. My only concern left is still the relatively small improvements over the baselines, with most increases being smaller than 1 AP point. I have seen the experiment about the statistical significance in A4.1, and this will obviously be considered in the final decision making.
> >
> > * Just to confirm, you did use a different random seed for each of the runs, correct?

---

> > > ### Author Response · Authors · 2022-11-29
> > > **Further discussion**
> > >
> > > Dear Reviewer EXB7,
> > >
> > > Thanks for your discussion. Our responses to your remaining concerns are summarized as follows.
> > >
> > > **Q9: Relatively small improvements (< 1 AP) over the baselines.**
> > >
> > > **A9:** On COCO dataset, it is difficult to get large improvements (all previous KD methods listed in our Table 1 have < 1 AP improvements compared to their prior methods), and we believe our improvements compared to previous methods is significant. For example, in Table 1, previous SOTA method FGD only improves GID (previous SOTA method that FGD compared) by 0.2 AP on `Faster RCNN-R50` and 0.1 AP on `FCOS-R50`; while our MasKD outperforms FGD by 0.4 AP on `Faster RCNN-R50` and 0.5 AP on `FCOS-R50`.
> > >
> > > **Q10: Did different random seeds used for each of the runs?**
> > >
> > > **A10:** Yes. We train the mask tokens and students with different random seeds independently in each run.

---

### Official Review · Reviewer_mLef · 2022-10-24

**Confidence:** 3
**Correctness:** 3
**Technical Novelty And Significance:** 3
**Empirical Novelty And Significance:** Not applicable
**Recommendation:** 8

**Clarity, Quality, Novelty And Reproducibility:**

Clarity: Fair. (Though most parts are well-written, some parts are still confusing.)

Quality: Good.

Novelty: Good. (I didn't see this technique before)

Reproducibility: Fair. (Releasing the code can further improve the reproducibility)

**Strength And Weaknesses:**

Pros:

1. Figure 2 clearly illustrates the framework of the proposed method.

2. This paper is overall well-written and easy to follow.

3. This paper is novel. I have never seen similar works before.

Cons:

1. For equation (7), how the weights of each mask are computed? What is the loss? Why this would not collapse to trivial solutions like emphasizing the masks that are easy to learn?

2. In Tables 1 and 2, the improvement over FGD seems marginal (around 0.2-0.4), which may come from randomness.


**Summary Of The Paper:**

This work proposes a new distillation method that is more fine-grained than the previous bounding box distillation methods. It achieves performance improvement in both object detection and semantic segmentation.


**Summary Of The Review:**

Though I still have some concerns, I think this paper is novel and well-written. Therefore, I would lean toward acceptance.

---

> ### Author Response · Authors · 2022-11-18
> **Response to Reviewer mLef**
>
> Dear Reviewer mLef,
>
> Thanks for your efforts in reviewing our paper. Our responses addressing your comments and questions are as follows.
>
> **Q1: For equation (7), how the weights of each mask are computed? What is the loss? Why this would not collapse to trivial solutions like emphasizing the masks that are easy to learn?**
>
> **A1:**
> 1. We have provided detailed descriptions of the mask weighting module in A.2 of our revision. As shown in Figure 3 (a), the mask weighting module (yellow box) first takes distillation feature $\mathbf{F}^{(t)}$ as input, then conducts a sequential of $3\times 3$ convolution and average pooling to get the feature of the whole image, then a $1\times 1$ convolution is adopted to predict the importance scores of $T$ mask tokens.
> 2. The weighting module is optimized jointly with the mask tokens using $\mathcal{L}_\mathrm{token}$ (Eq. (6)).
> 3. Due to the fact that the weights $\mathrm{w}_i$ are optimized with task loss, those masks who contribute more to the precision would be assigned more attention instead of simply resorting to the difficulty of learning.
>
> ---
>
> **Q2: In Tables 1 and 2, the improvement over FGD seems marginal (around 0.2-0.4), which may come from randomness.**
>
> **A2:** Thank you for your concerns about our performance on object detection. In our method, we simply kept the same configurations of our distillation loss in all settings. While we find that the optimal hyper-parameters vary in different detectors, and the recent state-of-the-art FGD adopts quite different configurations (e.g., loss weight) on each detector. Therefore, we further conduct experiments on all the students in Table 6 with better configurations following FGD. The results are summarized as follows.
>
> |Teacher|Student|Method|AP|AP$_{50}$|AP$_{75}$|AP$_S$|AP$_M$|AP$_L$|
> |:--:|:--:|:--:|:--:|:--:|:--:|:--:|:--:|:--:|
> |FasterRCNN-R101|FasterRCNN-R50|FGD|40.4|-|-|22.8|44.5|53.5|
> |FasterRCNN-R101|FasterRCNN-R50|MasKD|40.8 (**+0.4**)|60.9|44.5|23.2|44.2|54.1|
> |Retinanet-R101	|RetinaNet-R50	|FGD	|39.6	|-	|-	|22.9	|43.7	|53.6|
> |Retinanet-R101	|RetinaNet-R50	|MasKD	|40.1 (**+0.5**)	|59.3	|42.8	|21.7	|43.9	|54.0|
> |FCOS-R101	|FCOS-R50	|FGD	|42.1	|-	|-	|27.0	|46.0	|54.6|
> |FCOS-R101	|FCOS-R50	|MasKD	|42.6 (**+0.5**)	|61.2	|46.3	|26.5	|46.9	|54.2|
> |CM RCNN-X101	|FasterRCNN-R50	|FGD	|42.0	|-	|-	|23.8	|46.4	|55.5|
> |CM RCNN-X101	|FasterRCNN-R50	|MasKD	|42.6 (**+0.6**)	|62.9	|46.3	|23.6	|46.6	|56.3|
> |RetinaNet-X101	|RetinaNet-R50	|FGD	|40.7	|-	|-	|22.9	|45.0	|54.7|
> |RetinaNet-X101	|RetinaNet-R50	|MasKD	|41.4 (**+0.7**)	|60.6	|44.1	|22.8	|45.7	|56.2|
> |Reppoints-X101	|Reppoints-R50	|FGD	|41.3	|-	|-	|24.5	|45.2	|54.0|
> |Reppoints-X101	|Reppoints-R50	|MasKD	|41.7 (**+0.4**)	|62.6	|45.0	|24.1	|45.8	|55.0|
>
> We believe the improvements are significant enough to show our superiority. Moreover, FGD is designed for object detection only (it leverages the ground-truth bounding boxes to compute the loss), while our MasKD is more generic and can achieve state-of-the-art performance consistently on object detection and semantic segmentation tasks.
>
> ---
>
> **Q3: Releasing the code can further improve the reproducibility.**
>
> **A3:** Thanks for your useful suggestion. We have uploaded our code including learning mask tokens and training students as the supplementary material. We hope the code can help readers better understand our method and reproduce our results.

---

> > ### Comment · Reviewer_mLef · 2022-11-19
> > **Raise the score**
> >
> > Thanks for the authors' response. My concerns have been resolved. I would raise my score.

---

### Official Review · Reviewer_q4W5 · 2022-10-25

**Confidence:** 2
**Clarity, Quality, Novelty And Reproducibility:** This work meets the bar of publicatio…
**Correctness:** 3
**Technical Novelty And Significance:** 3
**Empirical Novelty And Significance:** 3
**Recommendation:** 8

**Strength And Weaknesses:**

Strengths:
1. The idea is novel and interesting. The mask tokens learned by minimizing the task loss can generate finer and task-related distillation regions, which are demonstrated through the visualizations and experiments in the paper.

2. The proposed method is generic to various tasks, as it does not require ground-truth labels to select distillation regions as previous methods.

3. The experiments on detection and segmentation tasks show that, MasKD achieves significant improvements compared to previous state-of-the-art methods. Sufficient ablation studies are provided to show the efficacy of the method.

Weaknesses:
1. The performance on ImageNet dataset is not state-of-the-art.
2. How long does it take to learn the mask tokens for the teacher?
3. Why the authors learn the mask tokens for only 2000 iterations? How about training it for more iterations?

**Summary Of The Paper:**

This paper proposes a knowledge distillation method for dense prediction tasks such as object detection and semantic segmentation. The method proposes to learn important regions (masks) adaptively for each image with task loss and mask diversity loss. Then the learned mask tokens, as well as mask weighting module, are adopted to distill teacher features to student. Results on COCO and Cityscapes datasets show the effectiveness of the proposed method.


**Summary Of The Review:**

I don't find major issues with this work. Please see my comments in strengths and weakness.

---

> ### Author Response · Authors · 2022-11-18
> **Response to Reviewer q4W5**
>
> Dear Reviewer q4W5,
>
> We thank you for your valuable comments. Our responses to your comments and questions are listed as follows.
>
> **Q1: The performance on ImageNet dataset is not state-of-the-art.**
>
> **A1:** Thank you for your concerns about our performance on classification. We propose MasKD mainly for dense prediction tasks such as object detection and semantic segmentation. While on image classification,  the improvement of MasKD is limited since its pixel-level features are less informative and discriminative. However, our MasKD still surpasses the vanilla KD and improves mimic by 0.55%.
>
> ---
>
> **Q2: How long does it take to learn the mask tokens for the teacher?**
>
> **A2:** We train the mask tokens for only 2000 iterations and it takes ~20 minutes on 8 V100 GPUs. Compared to the ~24 hours’ training time of the distillation stage, we believe this mask learning time is negligible.
>
> ---
>
> **Q3: Why MasKD learns the mask tokens for only 2000 iterations? How about training for more iterations?**
>
> **A3:** Thanks for your valuable suggestions. We empirically find that the mask tokens can get converged within 2000 iterations. Besides, we further conduct experiments on different training iterations with Faster RCNN-R50 student and Cascade Mask RCNN-X101 teacher. As the results shown in the following table, with only 2000 iterations, our mask tokens can achieve saturated performance.
>
> |Iterations|0 (random init.)| 1000 | 2000 | 4000 | 8000 | 12000 |
> |:--:|:--:|:--:|:--:|:--:|:--:|:--:|
> |mAP|40.4|41.1|41.4|41.4|41.5|41.3|

---

### Decision · Program_Chairs · 2023-01-20

**Decision:**

Accept: poster

**Justification For Why Not Higher Score:**

This work remains to be an empirical idea and has no particularly exciting insight. As one reviewer pointed out during the discussion, the combination of the moderately creative technical idea and the rather minor improvements put a limitation on this paper's merit

**Justification For Why Not Lower Score:**

The distillation approach does not require supervision, and it gives a consistent (albeit small) improvement. The benefit of distillation without ground-truth labels in this paper is a more general improvement compared to the existing methods. The experimental gains are consistent even though not always significantly



**Metareview: Summary, Strengths And Weaknesses:**

This work proposes a new distillation method that is more fine-grained than the previous bounding box distillation methods. It generates masks in a task-agnostic way,  in the sense that no priors about the output space (bounding boxes or segmentation maps) are used. It achieves performance improvement in both object detection and semantic segmentation. Overall, the paper is well written.

Two out of the three reviewers recommend this paper highly favorably after rebuttal. One reviewer indicates that most of his/her prior concerns (experiment sufficiency, motivation clarity, etc.) have been addressed. The only concern left is still the relatively small improvements over the baselines, with most increases being smaller than 1 AP point. It has been noted that the experiment about the statistical significance in A4.1 seems to endorse the statistical significance and consistency of the performance gains.

After reading the paper and holding internal discussions, AC feels that the overall sentiment is positive about this paper, and the negative reviewer stated that he/she won't object to acceptance. Hence, AC recommends this paper to pass the bar, as a poster presentation.





**Note From Pc:**

if the above contains the word "oral" or "spotlight" please see: "oral" presentation means -> notable-top-5% and "spotlight" means -> notable-top-25%. As stated in our emails, we are disassociating presentation type from AC recommendations